# SKATE, A SCALABLE TOURNAMENT EVAL: WEAKER LLMS DIFFERENTIATE BETWEEN STRONGER ONES USING VERIFIABLE CHALLENGES

## ABSTRACT

Evaluating the capabilities and risks of frontier AI models is paramount, yet current methods demand extensive domain expertise, hindering their scalability as these models rapidly evolve. We introduce SKATE: a novel evaluation framework in which large language models (LLMs) compete by generating and solving verifiable tasks for one another. Our core insight is to treat evaluation as a game: models act as both task-setters and solvers, incentivized to create questions which highlight their own strengths while exposing others' weaknesses. SKATE offers several key advantages, balancing scalability, open-endedness, and objectivity. It is fully automated, data-free, and scalable, requiring no human input or domain expertise. By using verifiable tasks rather than LLM judges, scoring is objective. Unlike domain-limited programmatically-generated benchmarks (e.g. chess-playing or spatial reasoning), having LLMs creatively pose challenges enables open-ended and scalable evaluation. As a proof of concept, we introduce LLM-set code-output-prediction (COP) challenges as a verifiable and extensible framework in which to test our approach. Using a TrueSkill-based ranking system, we evaluate six frontier LLMs and find that: (1) weaker models can score stronger ones consistently, reliably differentiating between them, and (2) LLM-based systems are capable of self-preferencing behavior, generating questions that align with their own capabilities, and (3) SKATE automatically surfaces fine-grained capability differences between models. Our findings are an important step towards general, scalable evaluation frameworks which can keep pace with LLM progress.

## 1 INTRODUCTION

The extraordinary evolution of large language models (LLMs) underscores the need for robust, scalable, and unbiased evaluation methodologies. As these models are increasingly deployed in high-stakes settings - such as healthcare, education, legal reasoning, and autonomous decision-making - the risks of mistake or misuse become more pronounced (Weidinger et al., 2021; Bommasani et al., 2022; Perez et al., 2022a). Current evaluation frameworks, however, often fall short of these goals. They typically require extensive human expertise or qualitative assessments. Furthermore, they are often **costly** to develop and maintain, **not scalable** enough to keep pace with the advancements of new models, and can be **manipulable, static, or gameable** (Weng, 2024; METR, 2025).

To address these limitations, we introduce a novel, fully automated, scalable and data-free evaluation framework. In our setup, LLMs compete by defining and solving verifiable tasks for one another. The "LLMs-as-task-setter" approach allows the evaluation scope to dynamically scale with the increasing complexity of the models themselves. By casting evaluation as a competitive, multi-agent game, SKATE creates pressure for models to both expose one another's weaknesses and showcase their own strengths. This makes SKATE ideal for tracking subtle performance differences and emergent behavior among advanced models.

LLMs setting tasks for themselves inherently presents limitations, such as potential self-preferencing biases (Gera et al., 2025) or awareness of being tested (Perez et al., 2022b). To mitigate these issues,

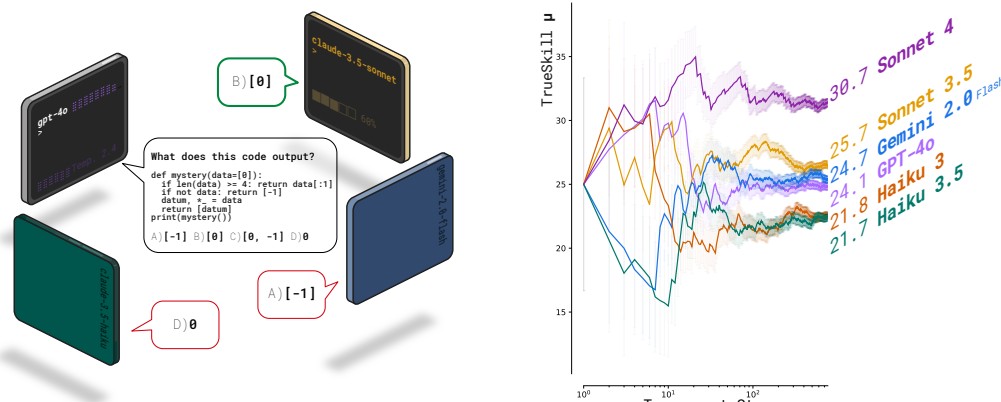

Figure 1: On the left: A Game of SKATE. A set of LLMs take turns to set questions for one another. Players are incentivized by their prompts to write questions which they can answer, but which their competitors cannot. In this way, the complexity of the generated questions scales with the capabilities of the setters themselves. On the right: the TrueSkill rank of each of six frontier models, based on their question answering ability, is initially uncertain and game outcomes are surprising. Eventually, a stable ranking emerges.

our framework is designed as a **peer-challenge game** where LLMs actively compete, creating a more objective and dynamic evaluation environment.

To realize this framework, we had to overcome several key challenges. These included mitigating multiple-choice related noise in LLM responses, and developing robust methods to measure task-setter success, focusing on metrics like question variability which we address through the development of a **similarity metric** for tasks.

As a proof of concept, we introduce LLM-set Code-Output-Prediction (COP) challenges. This verifiable and extensible framework serves as a concrete testbed for our approach. While this paper focuses on COP, superficially a narrow task, we claim that this approach has broad applicability: since computational tasks can be expressed as code execution problems, COP provides a general substrate for evaluating model capabilities, such as reasoning, across diverse domains.

We apply our peer-challenge game to six frontier LLMs, employing a TrueSkill (Herbrich et al., 2006)-based ranking system to quantitatively assess their performance.

We summarize our contributions below:

1. SKATE: an **automated**, and **scalable** evaluation framework where LLMs act as both task-setters and solvers in a competitive, verifiable game.

2. Evidence that **weaker models can reliably score stronger ones**.

3. Evidence that LLM-based systems exhibit a **capacity to write self-favoring questions**.

4. Automatic discovery of **differentiating questions**, exposing fine-grained capability differences between models.

## 2  RELATED WORK

**Benchmarks and evaluations**  Many benchmarks aim to evaluate the capabilities of AI models. Some are general purpose (Hendrycks et al., 2021a) or for general agentic tasks (Mialon et al., 2023), others for particular skills or knowledge areas (Li et al., 2024; Hendrycks et al., 2021b). Evaluations of coding performance range from single-completion generation of code (Chen et al., 2021), to agentic resolution of GitHub issues (Jimenez et al., 2024), to ML engineering tasks (Huang et al., 2024), and to many others. Criticism includes suggestions that benchmarks may be unrealistic (Kapoor et al., 2024; Becker et al., 2025), or distract from higher-priority safety interventions (Ren et al., 2024).

Evaluations often take significant effort to produce (Phan et al., 2025)), and those which are tuned to be sensitive to model performance at the point of publication are often quickly "saturated", as models progress faster than expected (Bengio et al., 2024) and reach indistinguishably-high performance (Hendrycks et al., 2021b). Our paper describes an approach where the generation of the evaluation is automated, and where a cohort of LLM peers can distinguish model capabilities, scaling as ever more capable models are developed.

**Verifiable games**  Arbitrary tasks can be scored using LLM judges, but expose the evaluation to the judges' biases (Gera et al., 2025). Game-based evaluations provide advantages including objective outcomes, verifiable moves, reduced data contamination (representation in training data), and scalability. Even simple grid-based games such as Tic-Tac-Toe and Connect Four (Topsakal et al., 2024) reveal variations in LLM performance across different games and prompt types. Chess has also been used as a testbed (Zhang et al., 2025; Wang et al., 2024a; Diallo & Akhloufi, 2025), including a leaderboard against random play (Saplin, 2024), showing that both fine-tuning and language explanations can enhance the performance and reasoning capability of large language models, as can tools such as RAG. In all settings, the ultimate goal is to find strategic planning and decision-making abilities which generalize to real-world tasks.

There is existing work involving evaluating LLMs in head-to-head style games. In Duan et al. (2024) LLMs are compared via their head-to-head performance on an environment containing 10 games. In Alyahya et al. (2025) LLMs are evaluated against one another in a competition-based framework with 7 static types of game. An important distinction between this literature and our work is that both papers' tasks are all human-crafted, whereas in SKATE, LLMs themselves design and pose all the questions in a scalable and dynamic setting.

**Automated task-setting**  The idea of LLMs generating tasks has been explored in prior work. Lu et al. (2025) focus on high-level capabilities and qualitative comparisons between models; crucially, their tasks are *open-ended* and validated using LLMs-as-judges. We propose to generate **objective, automatically verifiable** tasks, and provide a scalable mechanism to use these tasks to rank models. Perez et al. (2022b) have used LLM-generated datasets to automatically probe a broad range of nuanced properties such as sycophancy, and concerning goals such as resource acquisition and goal preservation. However, while the authors find that crowdworkers agree with 90–100% of labels, the questions set by LLMs are not automatically verifiable and it is unclear whether the dataset-generation approach fully avoids the pitfalls of LLM-judges Gera et al. (2025). Other work generates tasks with templates (Weston et al., 2015) or programmatically (Johnson et al., 2016), which, while verifiable, limit the diversity, expressivity and scalability seen in our model-written COP approach.

**Automated red-teaming**  Automated Red Teaming (ART) finds cases where a target LLM behaves in a harmful way, by generating test cases ("red teaming") using another LLM (Perez et al., 2022a). Active research has led to many advances, including curiosity-driven exploration (Hong et al., 2024), structured search (Mehrotra et al., 2024), reinforcement learning, prompt engineering and optimization, and transferability (Raheja et al., 2024). However, to our knowledge these approaches focus exclusively on jail-breaking, rather than an open-ended exploration of the limits of a model's capabilities.

**Scalable oversight**  Concerned by the potential future need to supervise systems which broadly outperform humans, and which are hard to evaluate, various groups have researched methods for weaker models to elicit the capabilities of stronger ones (Burns et al., 2023; Bowman et al., 2022). While these methods constitute progress towards aligning superhuman models to the goals of weaker models, with SKATE weaker models can evaluate the *capabilities* of stronger ones, to choose which strong student to teach.

## 3  ~~BACKGROUND~~ SETTING AND SCORING VERIFIABLE TASKS

Our goal is to create a fully-automated evaluation framework for LLMs. While highly desirable for its scalability, full automation presents significant challenges, particularly in ensuring the trustworthiness and reliability of the evaluation process. Using LLMs as question generators raises immediate concerns about the verifiability of generated questions and the potential for judge bias.

Our solution hinges on using **verifiable tasks**, implementing a **robust scoring mechanism**, and employing **question clustering** to prevent redundancy. We address each of these in turn below.

**LLMs as Setters of Verifiable Tasks**    The idea of using LLMs to evaluate other LLMs is appealing, but unconstrained ~~question setting~~ question-setting often yields questions that are subjective or costly to verify (e.g. "write a moving poem", or complex legal queries). Whilst expensive verification can be mitigated by "LLMs-as-judges", this approach risks inherent biases (Gera et al., 2025). To counter this, we design our framework around **verifiable tasks** - those with clear, systematic, and objective assessable solutions.

**Code-Output Prediction as a General Testbed**    For the purposes of our work, we ~~choose~~ experiment using Code-Output-Prediction (COP) tasks ~~as our testbed~~: *given a block of code, what does the code output?* Correct answers can be determined by via a code-execution sandbox. In Appendix C we show that many types of cognitive tasks, such as "counting the number of 'r's in "strawberry", mathematical and spatial reasoning, and various games e.g. chess, can be converted into COP tasks.

Other examples of verifiable tasks include games, writing code to pass unit tests, and factual questions with definitive answers ~~. We bound the output space of model answers, to quantify a model's performance on a question given the probabilistic nature of LLMs, by making our tasks~~ (see Appendix B). For SKATE to be able to use a verifiable task, it must be practical to automatically determine that task's ground truth answer. In this work, we choose to restrict tasks to be multiple-choice ~~.~~ but emphasize that the framework does not require this[1]. "Distractor" incorrect options are also generated by the question-asking model, while the correct answer is determined using a code-execution sandbox. While our experiments focus on COP, the SKATE framework is **general-purpose** and can be instantiated with any such verifiable task type. We note, however, that ~~our~~ restricting to a multiple-choice formulation in our experiments inherently restricts the evaluation space, for example by excluding open-ended responses, which may capture different dimensions of model capability.

**Robust Scoring for MCQs**    LLM responses to multiple choice questions (MCQs) are sensitive to factors like option ordering (Zheng et al., 2024) and option content, see Figure~ 6. To account for this, we estimate a model's score on a question by sampling it multiple times with randomly permuted answer sets. Each includes the correct answer and three distractors selected from a pool of nine. We repeat this process until the estimated accuracy, denoted **p(correct)**, has standard deviation below a stability threshold ($\sigma^* = 0.05$). The *choice of $\sigma^*$* reflects a balance: too large a value would make model comparisons uninformative, while too small a value would be computationally costly. This yields robust per-question probabilities $\in [0, 1]$ that we use to compute both relative and absolute rankings of models (see Section 4.3 and Appendix A).

**Question Clustering**    Measuring the similarity of generated tasks is crucial for categorizing capabilities and preventing models from "reward hacking" through repeated, similar questions. We aim to incentivise exploration of diverse capabilities. To this end, we use vector embeddings (specifically `openai-text-embedding-3-small` (Neelakantan et al., 2022))[2] and define question similarity based on a cosine similarity distance threshold. From empirical analysis of a diverse dataset of 2,977 questions, we found a threshold on $d_{ij} = 1 - d_{ij}^{\cos}$ of $d_{\text{thresh}} = 0.336$ effectively groups distinct questions [3]. More information is in Appendix D.

---

[1]In Appendix B we give some examples of different types of COP which are unsuitable for multiple-choice: where automatically generating ground-truth answers is difficult but automatically verifying solutions is trivial (e.g. theorem proving in a formal language).

[2]It is possible that using different embedding models could lead to different filtering decisions, but we expect the overall impact of this to be minimal for current models, especially since we observe empirically that the filtering step mostly removes fairly obvious near-duplicates.

[3]This threshold is a design choice: from empirical observation it separates questions into distinct clusters amongst the models we tested, but may need adjustment for newer models.

# 4 PEER-CHALLENGE LLM BENCHMARK GAME

In this section we define the format of our "Game of SKATE", and how it is scored. With this in place, we define various "augmentation strategies" which allow models to take advantage of various pieces of information (e.g. models' p(correct) on previous questions). Most evaluation frameworks treat models as passive subjects: evaluated via curated benchmarks or third-party judges. In contrast, SKATE frames evaluation as an adversarial game: models actively participate as both task designers and solvers.

## 4.1 DEFINING THE GAME

$N$ LLMs take turns asking and answering questions for $N_{\text{rounds}} = 50$. In each round, each player asks one question of all other players (and must also attempt to answer the question itself). Players have $N_{\text{attempts}} = 3$ attempts to create a suitably verifiable and distractor-rich question $q$, defined as:

- **Verifiable:** The question has a unique and assessable answer. For COP this means $q$ runs without error in a code-execution sandbox.

- **Distractor-rich:** The player successfully generates 9 unique incorrect "Distractor" options for the question $q$.

We refer to questions which are Verifiable and Distractor-rich as **valid**. Lastly, for a player to enter the round, we require this valid question to be suitably unique as defined below:

- **Unique:** distance$(q, q_j) > d_{\text{thresh}}$ for all questions $q_j$ previously set by the player.

The uniqueness criterion forces the Asking model to prioritize question variability and avoids the risk of them fixating on one particular capability or question style.

We therefore have a set of questions for the round $\{q_1, q_2, ..., q_N\}$, some of which may be empty if the player fails to create a unique, valid question. All players attempt to answer all questions, and are scored using the the algorithm outlined in Section A. Note that at each "attempt" within a round, the player has access to its previous attempts for that round and why they failed (i.e. not verifiable, not enough distractor options, or not "unique enough").

It is possible to give all players access to full information about the game state in each round, including all previous questions asked by all players (including themselves), and the p(correct) scores achieved by all players on those questions. However, since this represents a large amount of information, in Section E we propose a variety of **augmentation strategies** which permit the task-setters to ingest increasing amounts of this information in the context window during task setting. Concretely: we draw a distinction between the **game rules** which in principle allow full access to game state information, and **augmentation strategies** which involve players using some, or all, of this information to help them play optimally.

## 4.2 INCENTIVES

At the level of the prompt, we tell the task-setting LLMs that they are playing a game and will be rewarded as follows (see Appendix J for the full prompt): a) +1 if it successfully created a **valid** question and, b) +1 for each question it gets correct (for both its own q and those of its competitors). We emphasize that these rewards are *not* the scoring mechanism we use to rank the models (which we detail in Section 4.3). but rather information used only in the prompt, designed to incentivize task-setters to write **valid questions** which **they can answer** but their **opponents cannot**.

We refer to these optimal questions as **discriminatory questions**, defined as valid questions which the task-setter answers correctly whilst the competitors answer *incorrectly*. Our game is designed such that optimal behavior for a player is to write 50 discriminatory questions. By requiring suitably *unique* questions using the similarity metric, we require the players to isolate as many **discriminatory niches** as possible; that is, to maximise the number of distinct areas in which they excel and their competitors do not.

## 4.3 RANKING

We use a *ranking system* to order models after the game. We employ TrueSkill (Herbrich et al., 2006), which is a Bayesian ranking system designed to rate the skill of players in a competitive game whilst taking into account uncertainty of their skill. Instead of a single skill score, TrueSkill represents a player's skill as a probability distribution, characterized by a mean $\mu$ and a variance $\sigma^2$.

We initialize all players with default starting skill $\mu = 25$ and level of uncertainty $\sigma = 25/3$. The TrueSkill algorithm uses Bayesian inference to update players' skills and uncertainties after obtaining new information about their competitive performance. At a coarse level: TrueSkill reduces uncertainty $\sigma$ for players whose performance aligns with their current skill estimate, and adjusts their mean skill $\mu$ upwards or downwards based on whether they over- or under-performed against expectations.

We present two ways to use MCQ answers to inform TrueSkill rankings:

1. **Relative Pairwise**: We define a win/loss/draw between two players based on the difference $\delta = |p_1 - p_2|$ between the p(correct) values they achieve. If this difference is less than $\sigma^* = 0.05$ (the model scoring stability threshold) then the models draw, else the higher-scoring model wins. TrueSkill is updated multiple times per round: once per unique pair of players.

2. **Absolute Pairwise**: We define a win/loss/draw between two players by marking each player as pass/fail based only on their own p(correct) value relative to a threshold $p_{\text{thresh}} = 0.55$. For example, if model gets p(correct)= 0.8 and another gets p(correct)= 0.6, then because both are above the threshold we would mark this a draw. TrueSkill is updated multiple times per round: once per unique pair of players.

## 5 AUGMENTATION STRATEGIES

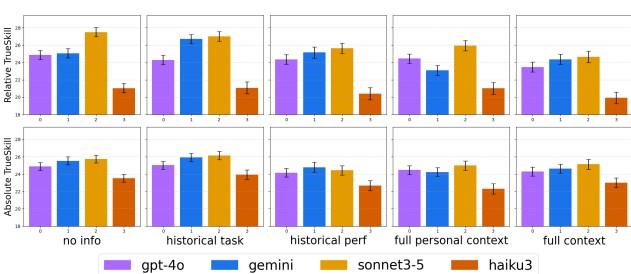

Figure 2: TrueSkill $(\mu, \sigma)$ for four models after one iteration of our game. We compare five augmentation strategies, and score each game using *relative* TrueSkill (top row) and *absolute* TrueSkill (bottom row). *Relative* TrueSkill produces clearer separation between models, reflecting its higher information content. The values quoted are the average values over the final 100 update steps.

We propose that sufficiently powerful LLMs could use game data (e.g. questions, and p(correct) values) to optimally construct future questions which favor themselves and disadvantage other players. To test this, we design five "augmentation strategies" which can be viewed as filters which permit types of game-state information to enter the task setters' contexts. These strategies range from "no-info" (no context about previous questions or performance), up to "full-context" (full information about all previous questions and all players' p(correct) values on those questions). These five strategies are described in detail in Appendix E and Table 1.

In Figure 2, we present results from five iterations of SKATE, each involving four players (GPT-4o, Sonnet 3.5, Haiku 3 and Gemini-2.0-Flash) equipped with one of five distinct augmentation strategies. While these strategies provide varying levels of access to the historical game state, we observe that they have minimal effect on the final rankings. Sonnet 3.5 consistently emerges as the highest-ranked model, and Haiku 3 as the lowest.

However, the augmentation strategies do influence how models generate questions over time. We show in Appendix E that some strategies lead to more adaptive question setting, with models refining difficulty to match their own abilities and better challenge others. This suggests that the models

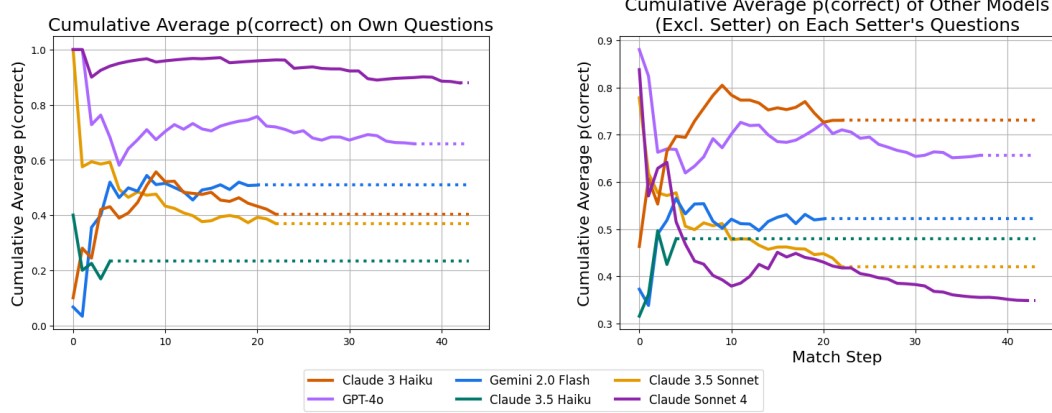

Figure 3: Cumulative average p(correct) values per model. Lines are different lengths depending on how many valid, unique COP questions each model was able to create in the 50 rounds.

tested can make limited use of game context information, but do not yet exhibit strong strategic exploitation of that information to be maximally adversarial.

To strike a balance between informativeness and computational cost, we adopt the **historical performance** augmentation strategy for the rest of our experiments. Whilst we expect that as LLM capabilities improve, models will better leverage full-context augmentation to construct highly targeted, adversarial tasks, our experiments cannot formally guarantee this. All experiments use temperature$= 0.7$.

## 6 A SCALABLE, AUTOMATED EVALUATION FRAMEWORK

We now run a full game with six players (Claude Haiku 3.0, Claude Haiku 3.5, Claude Sonnet 3.5, Claude Sonnet 4, Gemini 2.0 Flash and GPT-4o), and pick the historical performance augmentation strategy.

**Game Results**    On the right-hand side of Figure 1 we present the evolution of the TrueSkill $(\mu, \sigma)$ values for each player throughout the game (using *relative* scoring). There is initial high variance as most game results are *surprising* from the perspective of the uniform skill prior. Sonnet 4 outperforms all models considerably, whilst amongst the others there is both differentiation and degeneracy.

In Appendix H we present an alternative way of measuring the differences between the players. In particular, in Figure 12 we plot the *question setting* skill against the *question answering* skill of players: separating the task-setter's ability to write challenging questions for its competitors from the ability to answer its competitors questions. It is interesting to note that most models' skills are equally balanced, but Sonnet 4 and Haiku 3.5 have skewed capabilities in opposing directions.

**Adaptive Question Setting**    A natural question to ask is whether *models adapt their question setting over time* based on the data they are given. In Figure 3 we analyse how the models' questions progress throughout the game. In the top panel we plot the cumulative average p(correct) score achieved by models on their own questions. We observe that GPT-4o, Sonnet 3.5 and Sonnet 4 begin by pitching questions which are *too easy* for themselves. On the other hand, the other three models initially pitch questions which they cannot answer. In most cases the cumulative score moves closer to the ideal middle ground: questions in the "sweet spot" which are as difficult as possible but still answerable by the task setter. A separate measure of question progression is presented in the bottom panel. Here, we plot the cumulative average of *other models'* p(correct) values on a task-setter's questions. Notice that both Sonnet 3.5 and 4 are learning to write progressively *more challenging* questions. Whilst Sonnet 4's own p(correct) values plateau, it is evidently still increasing the difficulty of the questions as measured from the perspective of its competitors.

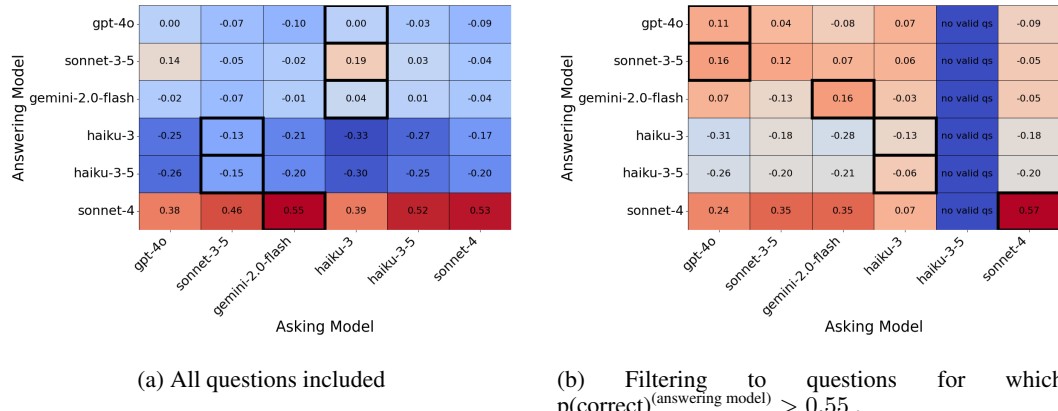

(a) All questions included

(b) Filtering to questions for which $p(correct)^{(answering\ model)} > 0.55$ .

Figure 4: Difference in average p(correct) scores between answering model and all other players. Positive values imply a model scores higher on average than its competitors. Highlighted cells are the maxima in each row. In (b) we observe close to maximal entries along the diagonal: with the filter in place, *models perform best on their own questions*. Note that there are "no valid questions" for Haiku 3.5 after applying the filter: it fails to write any questions which it can answer sufficiently well.

**Self-Preferencing**   In Figure 4 we study to what extent models are capable of *self-preferencing*: writing questions which favor their own capabilities over those of their competitors. We separate the question-set into tranches based on the task-setter. In Figure 4a, for each tranche of questions we plot the mean difference in p(correct) scores achieved by each player compared to their competitors. In Figure 4b we filter each tranche to only include questions on which the task-setter scored p(correct)> 0.55, corresponding to a task-setting system which checks it can answer its own challenge before posing it to others. We observe in the latter plot that after applying this filter, *all* models exhibit the capacity to create self-preferencing questions: GPT-4o, Gemini 2.0 Flash, Haiku 3 and Sonnet 4 all perform optimally on their own questions compared to average, and Sonnet 3.5 comes close.

**Comparison to Other Benchmarks**   In comparisons to other existing benchmarks (see Appendix I) we see moderate-to-strong correlation.

## 6.1 WEAKER MODELS CAN SCORE STRONGER MODELS

In Figure 5 we demonstrate that a collection of weaker agents are able to reliably score and differentiate two stronger models. Initially, we play a Game of SKATE between four weaker models (GPT-4o, Gemini-2.0-Flash, Haiku 3.0 and Haiku 3-5). Using this set of questions, we rank the four models in the first panel. In the second panel, we have Sonnet 3.5 and Sonnet 4 answer this set of questions - and re-rank all models. We find that the rankings are *stable* to the addition of the new models, and the existing question set suitably differentiates the two new, stronger, models. For

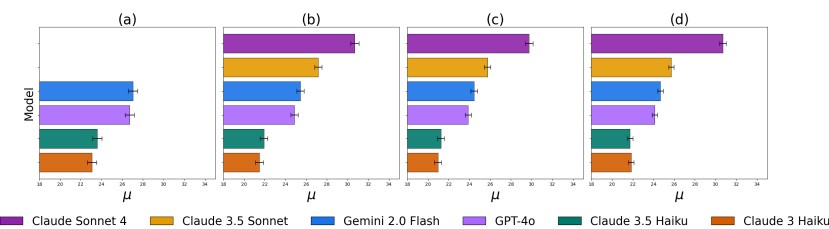

Figure 5: In (a), four "weaker" agents play a Game of SKATE. In panel (b), we use *questions from these four models* to rank two new "stronger" models (Sonnet 3.5 and Sonnet 4). In panel (c), Sonnet 3.5 joins the Game and sets its own questions, which all six models answer. In panel (d) Sonnet 4.0 also joins the Game and sets questions of its own.

completeness, in the third and fourth panels we have Sonnet 3.5 and Sonnet 4 join the game (and set their own questions). Notice that the rankings amongst the four panels are stable and relatively unchanged: the weaker models were able to reliably differentiate the stronger models.

## 6.2 RANKING IS STABLE TO ADDING NEW MODELS

Any scalable evaluation framework must be stable to the addition of new, more powerful, models. To test this, we experiment with adding models into the game sequentially, and with different ordering. We show that adding Sonnet 3.5, and then Sonnet 4 (and vice versa) preserves the relative TrueSkill score values of the existing models. See Appendix F for more details.

## 6.3 POINTS OF DIFFERENCE BETWEEN MODELS

Our framework is capable of surfacing questions which differentiate between models automatically, without human annotation or task curation, making it particularly suited for tracking subtle and evolving capability differences across model families. Questions with high p(correct) variance across models are those which models answer with varying confidence. In Appendix G we provide example questions of different p(correct) variance. For example, we observe a question which reveals a failure mode specific to certain models that is not obvious from the overall accuracy or ranking metrics: whilst Gemini 2.0 Flash is ranked *lower* than Sonnet 3.5, on this question it achieves p(correct)$= 1.0$ compared to Sonnet 3.5's p(correct)$= 0.0$. We observe interesting variants of strengths and weaknesses across all models. While a full taxonomy of model-specific strengths is beyond the scope of this paper, these examples serve as a proof of principle that *peer-generated* verifiable tasks can uncover meaningful and fine-grained performance differences. Importantly, these differences arise naturally through competitive dynamics without needing hand-crafted benchmarks or manually constructed capability categories.

## 7 DISCUSSION

This paper introduces a novel advancement in the evaluation of LLMs through the framework of "LLM-as-setter-of-verifiable-tasks". SKATE addresses critical limitations of current evaluation methodologies, which often demand significant human expertise, are costly, and lack scalability. By enabling LLMs to autonomously generate verifiable tasks for themselves and other LLMs, it offers a scalable evaluation solution capable of keeping pace with the progression of LLM capabilities. A core innovation of our work lies in the design of a peer-challenge game, which provides a dynamic evaluation environment. The use of verifiable tasks is central to ensuring objective and assessable solutions. While COP serves as a concrete testbed, the underlying principle of computational tasks expressed as code execution problems affords broad applicability across diverse domains.

Our experimental findings highlight several important aspects of LLM behavior and the efficacy of our framework. The TrueSkill-based ranking system proved effective in quantitatively assessing performance, revealing a stable hierarchy among the tested frontier LLMs. Interestingly, our investigation into augmentation strategies, designed to allow LLMs to leverage game state information, showed minimal impact on the rankings and small impact on adaptive question setting. This suggests that while the framework provides the necessary information, the tested models may not yet possess the advanced strategic reasoning capabilities to fully exploit it for optimal task construction. However, we anticipate that as LLM capabilities mature, they will increasingly utilize such information to design more advantageous questions.

In our six-player game experiment, the observed adaptive question setting behavior of LLMs is particularly noteworthy. Specifically, Sonnet 3.5 and Sonnet 4 demonstrated a capacity to write progressively more challenging questions, as measured by their competitiors' decreasing p(correct) values, indicating a sophisticated understanding of the competitive dynamics. A significant finding is the demonstration of self-preferencing, where all models, when filtered to questions they answered correctly, exhibited a capacity to design questions that favored their own capabilities over competitors.

Crucially, our framework proved its scalability and robustness by demonstrating that cohorts of weaker agents can reliably score and differentiate between unseen stronger agents. The stability of

rankings, even with the sequential addition of new, more powerful, models underscores the framework's ability to maintain transitive scoring relationships and effectively monitor LLM progress without requiring a constant re-calibration of benchmarks. This finding is key for developing general, scalable evaluation frameworks that can truly keep pace with the advancement of LLM capabilities. We also demonstrated that SKATE isolates *points of difference* between models - where p(correct) variance is high. As model capabilities continue to diversify, the ability to expose such differences scalably and automatically will become increasing valuable for safety, deployment and capability evaluation.

Whilst our similarity metric avoids redundancy in question setting, it does not formally guarantee diversity. It is possible that all LLMs in a game of SKATE share a common blind spot for COP which is not identified by any task setters. In future work one could consider seeding task setters with different questions generated by a range of models on a variety of tasks to promote further diversity.

A key limitation is that our prompting strategy may not reliably elicit the intended game-playing behavior, so negative results could reflect prompt design rather than true model failure. Models with code-execution abilities (e.g. tool-augmented LLMs) trivially solve our tasks, meaning COP is best suited for *pure* language models. Still, the SKATE framework is general: future work could couple COP with external information sources (e.g. physical world simulations or APIs) or add other verifiable tasks to mitigate this limitation. Our experiments also cover a narrow set of models, so extending SKATE to a ~~winder~~ wider range could expose richer capability patterns and provider-specific strengths. Finally, while COP is broadly applicable, it does not capture the full range of LLM capabilities, underscoring the value of exploring additional verifiable tasks. Our work, and the use of COP, serves a proof-of-principle of the SKATE approach ~~would~~ which could be applied more generally.

## 8 CONCLUSION

Rapid progress in AI capabilities requires rigorous evaluations which can keep pace, automatically spanning and detailing broad capability profiles. SKATE provides a framework for objective such evaluations which are robust to the introduction of increasingly capable models. This work also lays groundwork for visibility of strategic behaviours which may emerge as model capabilities improve, such as self-preferencing, metagaming (e.g. early sandbagging), and adaptive question setting.

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

## A    MC Scoring Algorithm

Multiple-choice question-answering (MCQA) is sensitive to various effects, see Figure 6.

Here we present the algorithm implemented to compute a model's score on a particular question, see Algorithm 1. We call this score the **p(correct)**. To be cost-effective, we adaptively sample the LLM for responses. For each response, we generate a varying set of four multiple-choice answers. The algorithm terminates when the standard deviation of the p(correct) is less than a threshold $\sigma \leq \sigma^*$. For this work, we choose $\sigma^* = 0.05$.

## B    Comments on Other Verifiable Tasks

In this section we discuss alternative verifiable tasks to COP which could also be implemented within the SKATE framework. We have argued (see Appendix C for more detail) that COP offers a general task setting in which a wide range of capabilities can be tested. However, further breadth can be explored by implementing entirely different classes of verifiable task.

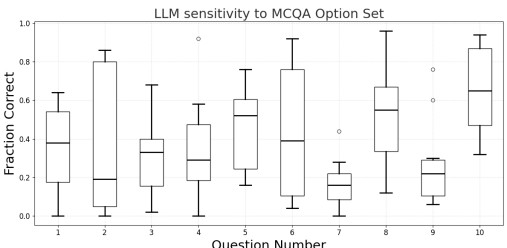 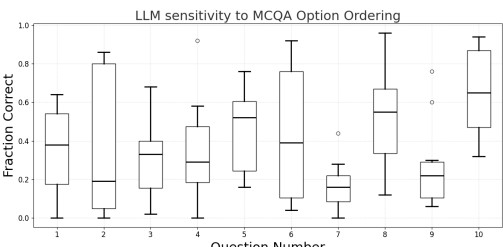

(a) Sonnet 3.5 sensitivity to MCQA option set on 10 questions. For each question, 10 sets of 4 MCQA options (including the ground truth answer) are posed to the Answering model. We measure "fraction correct" by sampling the LLM 10 times for each set of 4.

(b) Sonnet 3.5 sensitivity to MCQA option ordering on 10 questions. For each question, 10 variations of MCQA option ordering are posed to the Answering model. We measure "fraction correct" by sampling the LLM 10 times for each set of 4.

Figure 6: MCQA choice and ordering effects. Both the option set and its ordering have a large influence on the measured LLM "correctness". To account for this, we propose Algorithm 1.

---

**Algorithm 1** Estimating Accuracy with Distractors

---

1: **Input:** Question $q$ with ground truth answer $a^*$, set of 9 distractor answers $\{d_1, \ldots, d_9\}$
2: **Initialize:** $N \leftarrow 0, C \leftarrow 0$          $\triangleright$ $N$ is total question variants shown, $C$ is correct answers
3: **repeat**
4:     **for** $i = 1$ to $N_{\text{step}}$ **do**
5:          Randomly select 3 distractors $\{d_{i_1}, d_{i_2}, d_{i_3}\}$ from $\{d_1, \ldots, d_9\}$
6:          $O \leftarrow \text{Shuffle}(\{a^*, d_{i_1}, d_{i_2}, d_{i_3}\})$
7:          Present options $O$ to the model and record answer
8:          **if** model selects $a^*$ **then**
9:              $C \leftarrow C + 1$
10:         **end if**
11:         $N \leftarrow N + 1$
12:     **end for**
13:     $p \leftarrow C/N$
14:     $\text{std} \leftarrow \sqrt{\frac{1}{N} \cdot p \cdot (1-p)}$
15: **until** $\text{std} \leq \sigma^*$
16: **Output:** Estimated accuracy $p$ with uncertainty std

---

Below we give two concrete examples (mathematical reasoning in a formal language, and unit-test driven code generation). One could also consider examples such as: predicting optimal next moves in perfect-information games (e.g. chess), or predicting the state after $N$ steps in a determinstic physics simulation (e.g. many-body, fluid, orbital).

## B.1 MATHEMATICAL REASONING IN FORMAL LANGUAGE

Given a theorem in a formal language (e.g. Lean Moura & Ullrich (2021)) the answering model must provide a proof in the same formal language. Lean allows this to be automatically verifiable. A simple example would be *Theorem: For any natural numbers $a$ and $b$, if $a$ divides $b$ and $b$ divides $a$, then $a = b$.* Converted to formal language, this is:

```
-- Fill in a Lean proof of the following:
theorem dvd_antisymm {a b : Nat} (h1 :  a | b) (h2 : b | a) : a = b := by
```

And here is a solution provided by GPT-5:

```
theorem dvd_antisymm {a b : Nat} (h1 :  a | b) (h2 :  b | a) : a = b := by
  cases h1 with k hk ->
  cases h2 with l hl ->
```

```
have h : a * k = a * l := by
  calc
    a * k = b := by simpa using hk
    _     = a * l := by simpa using hl
have hk_eq : k = l := by
  have := Nat.mul_right_cancel a h
  simpa using this
have : b = a * k := hk
simpa [hk_eq, this]
```

This proof fails to compile as valid Lean code (the authors are not Lean experts but believe that a positivity proof on $a$ is required, e.g. an additional proof that $a > 0$), and so GPT-5 would fail this task. It is important to note that because it is impossible to automatically generate ground-truth solutions, this task family could not be posed as multiple-choice questions. Instead, one would generate a p(correct) score by sampling a model's response many times on this question and measuring a fraction of correct responses. For efficiency, to avoid the setting of unsolvable tasks, it may be desired to only ask questions for which the task-setting model can provide valid solutions, though this is not strictly necessary since TrueScore is not affected by tasks at which models perform equally well.

## B.2 UNIT-TEST DRIVEN CODE GENERATION

Within the domain of programming, an interesting type of test could be *unit-test driven code generation*. Task setters in this game would set a suite of test cases, and the answering model must write code to pass them.

Below we give an example of a verifiable code generation task produced by Gemini-2.5-pro:

```
import unittest

# The function to be implemented by the challenger model will be imported here.
# A placeholder is defined to allow the test suite to be self-contained.
def format_string(value, format_spec):
    """
    Formats a value according to a custom format specifier, without using any
    built-in formatting tools or external libraries.
    """
    raise NotImplementedError("""The 'format_string' function
    is not yet implemented.""")

class TestCustomStringFormatter(unittest.TestCase):

    def test_default_left_alignment(self):
        self.assertEqual(format_string("test", "10"), "test      ")
        self.assertEqual(format_string(123, "5"), "123  ")
        self.assertEqual(format_string("abc", "3"), "abc")

    def test_explicit_left_alignment(self):
        self.assertEqual(format_string("left", "<10"), "left      ")
        self.assertEqual(format_string("text", "_<8"), "text____")
        self.assertEqual(format_string(987, "-<5"), "987--")

    def test_right_alignment(self):
        self.assertEqual(format_string("right", ">10"), "     right")
        self.assertEqual(format_string("text", "_>8"), "____text")
        self.assertEqual(format_string(12, "0>4"), "0012")

    def test_center_alignment(self):
        self.assertEqual(format_string("center", "^10"), "  center  ")
```

```
1026                    self.assertEqual(format_string("text", "*^8"), "**text**")
1027                    # Test for uneven padding distribution
1028                    self.assertEqual(format_string("odd", "^6"), " odd  ")
1029                    self.assertEqual(format_string("odd", "*^6"), "*odd**")
1030
1031            def test_string_truncation(self):
1032                    self.assertEqual(format_string("this is a long string", "5"), "this ")
1033                    self.assertEqual(format_string("truncate", "<5"), "trunc")
1034                    self.assertEqual(format_string("truncate", ">5"), "trunc")
1035                    self.assertEqual(format_string("truncate", "^5"), "trunc")
1036                    self.assertEqual(format_string("exact", "5"), "exact")
1037
1038            def test_custom_fill_character(self):
1039                    self.assertEqual(format_string("fill", "-<10"), "fill------")
1040                    self.assertEqual(format_string("fill", "->10"), "------fill")
1041                    self.assertEqual(format_string("fill", "-^10"), "---fill---")
1042                    self.assertEqual(format_string(42, "0>5"), "00042")
1043                    self.assertEqual(format_string("pad", " ^6"), " pad  ")
1044
1045            def test_edge_cases(self):
1046                    self.assertEqual(format_string("", "10"), "          ")
1047                    self.assertEqual(format_string("a", "1"), "a")
1048                    self.assertEqual(format_string("a", "^1"), "a")
1049                    self.assertEqual(format_string(None, "<10"), "None      ")
1050                    self.assertEqual(format_string(3.14, ">6"), "  3.14")
1051                    self.assertEqual(format_string("a", "^2"), "a ")
1052
1053            def test_invalid_format_specifiers(self):
1054                    with self.assertRaises(ValueError):
1055                        format_string("test", "")  # Empty format spec
1056                    with self.assertRaises(ValueError):
1057                        format_string("test", "abc")  # No valid width
1058                    with self.assertRaises(ValueError):
1059                        format_string("test", "<>")  # Multiple alignment characters
1060                    with self.assertRaises(ValueError):
1061                        format_string("test", "x10")  # Invalid alignment character
1062                    with self.assertRaises(ValueError):
1063                        format_string("test", ">>10")  # Fill cannot be an alignment char
1064                    with self.assertRaises(ValueError):
1065                        format_string("test", "10<")  # Width specified before alignment
1066                    with self.assertRaises(ValueError):
1067                        format_string("test", "-^")  # Missing width
1068                    with self.assertRaises(ValueError):
1069                        # Fill must be a single character
1070                        format_string("test", "longfill^10")
```

and a solution provided by GPT-5

```
    def format_string(value, format_spec):
        """
        Formats a value according to a custom format specifier.

        Supported grammar (roughly):
          [fill][align]width
        where:
          - fill is a single character (optional)
          - align is one of '<', '>', '^' (optional; default '<')
          - width is a required positive integer
```

```
Examples:
  "10"  -> width=10, default left align, fill=' '
  "<10" -> left align, fill=' '
  "_>8" -> fill='_', right align, width=8
  "0>5" -> fill='0', right align, width=5
"""
if not format_spec:
    raise ValueError("Empty format specifier")

ALIGN = {'<', '>', '^'}
fill = ' '
align = '<'  # default alignment
width_str = None

# If second char is an alignment character, first char is fill.
if len(format_spec) >= 2 and format_spec[1] in ALIGN:
    fill = format_spec[0]
    if fill in ALIGN:
        # fill cannot be an alignment char
        raise ValueError("Fill cannot be an alignment character")
    align = format_spec[1]
    width_str = format_spec[2:]
elif format_spec[0] in ALIGN:
    # First char is alignment, no explicit fill
    align = format_spec[0]
    width_str = format_spec[1:]
else:
    # No explicit fill/align at the start: treat entire spec as width
    width_str = format_spec

# Width must be present and numeric
if not width_str or not width_str.isdigit():
    raise ValueError("Invalid or missing width")

width = int(width_str)

s = str(value)

# Truncate if longer than width
if len(s) > width:
    return s[:width]

pad = width - len(s)
if align == '<':
    return s + (fill * pad)
elif align == '>':
    return (fill * pad) + s
elif align == '^':
    left = pad // 2
    right = pad - left
    return (fill * left) + s + (fill * right)
else:
    raise ValueError("Unknown alignment")
```

GPT-5's solution passes all the tests. Similar to the theorem-proving task above, this type of task is also unsuitable for the multiple-choice format but can naturally be posed as an open-ended question and scored via sampling. An interesting extension of this task type would be to allow the task

## C COMMENTS ON GENERALIZABILITY OF CODE OUTPUT PREDICTION TASKS

In this section we comment on the generalizability of COP tasks as a substrate for evaluating model capabilities. We aim to give examples of broad classes of problems which can be phrased within COP. In doing so, we demonstrate that SKATE with COP is broadly applicable to assessing a broad range of capabilities. We also emphasize that SKATE is not limited COP tasks, and discuss other types of verifiable tasks in Section B

Problems of the below styles would all be possible within the SKATE framework. Whilst our SKATE experiments found that models focused on algorithmic / programming reasoning when given free-reign on task setting, one could straightforwardly adapt the prompts to steer task setters towards setting other types of COP tasks.

Note that in all the examples below, the questions were both *ideated* and *translated into COP* by GPT-5 unless otherwise specified. This demonstrates that state-of-the-art models can in principle generate valid, interesting, and meaningfully-diverse questions of these types.

### C.1 CLASSIC BLIND SPOTS

We begin with a very simple example to emphasize the details of our claim. Early language models famously struggle to count the frequency of the letter 'r' in 'strawberry'. Questions in this style, isolating common failure modes of LLMs are easily re-framed in COP:

```
"strawberry".count("r")
"Mississippi".count("s")
"abracadabra"[2:7]
"hello world".index("o")
```

Crucially, each of these COP questions is automatically verifiable but still assesses the core competency (counting letters and indexing).

### C.2 MATHEMATICAL REASONING

We now move onto mathematical reasoning. A wide class of mathematical problems can be reformulated as COP tasks. Below we give some examples.

The first tests simple Boolean logic:

```
import itertools

def sat(A,B,C):
    return (A or B) and ((not A) or C)

for vals in itertools.product([False,True], repeat=3):
    if sat(*vals):
        print("Satisfiable with:", vals)
        break
else:
    print("Unsatisfiable")
```

To give some more complex examples, models could set questions which can be solved by brute force (such that they fit within the COP-framework), but humans would solve via convenient tricks. For example, consider $137^{25} \mod 143$, which can be solved with simple number theory tricks. This is simply converted to COP via:

```
print(pow(137, 25, 143))
```

Similarly, solutions to $x^2 - 5y^2 = 44$ can be found via brute force as a COP task

```
solutions = []
for x in range(-100, 101):
    for y in range(-100, 101):
        if x*x - 5*y*y == 44:
            solutions.append((x,y))

print(solutions)
```

but are most easily solved with advanced mathematical reasoning.

## C.3 SPATIAL REASONING

We now give three examples of spatial reasoning tasks. Firstly, a model could pose a COP task where it challenges its competitor's ability to reason about rotating a dice:

```
class Dice:
    def __init__(self):
        self.faces = {
            "top": "1", "bottom": "6",
            "front": "2", "back": "5",
            "left": "3", "right": "4"
        }

    def _rotate(self, plane, angle):
        """Some function to rotate the faces given a plane and angle"""
        self.faces = # update rules

    def get_face(face):
        return self.faces[face]

d = Dice()
d._rotate('x', 90)
d._rotate('y', 270)
print(d.get_face('top') + d.get_face('left'))
```

This pseudocode was written by the authors. Secondly, consider task asking *given a graph, can node A reach node B via a path?*. Humans could solve this easily using spatial reasoning, and this is what we want to test. The challenge can be converted to COP using path search methods:

```
from collections import deque

graph = {
    "A": ["B","C"],
    "B": ["D"],
    "C": ["E"],
    "D": ["F"],
    "E": [],
    "F": []
}

def reachable(start, goal):
    q = deque([start])
    seen = set()
    while q:
        u = q.popleft()
        if u == goal: return True
```

```
                for v in graph[u]:
                    if v not in seen:
                        seen.add(v)
                        q.append(v)
        return False

    print(reachable("A","F"))
```

Next, the task *is this 5-node graph 3-colorable?* is converted to COP as follows:

```
    import itertools

    edges = [(0,1),(1,2),(2,3),(3,4),(4,0)]
    colors = [0,1,2]

    for assignment in itertools.product(colors, repeat=5):
        if all(assignment[u] != assignment[v] for u,v in edges):
            print("3-colorable:", assignment)
            break
    else:
        print("Not 3-colorable")
```

Lastly, the task *How many paths are there from $(0, 0)$ to $(6, 6)$ that move only right or up, but don't pass through $(3, 3)$?* can be converted as follows:

```
    from itertools import product

    def count_paths(n):
        paths = 0
        # Represent a path as a sequence of 'R' and 'U'
        for seq in product("RU", repeat=2*n):
            if seq.count("R") == n and seq.count("U") == n:
                x = y = 0
                ok = True
                for move in seq:
                    if move == "R": x += 1
                    else: y += 1
                    if (x, y) == (3,3):
                        ok = False
                        break
                if ok:
                    paths += 1
        return paths

    print(count_paths(6))
```

## C.4 GAMES/ PUZZLES

We now consider two examples of games (connect-4 and chess). It is straightforward to convert interesting challenges within these settings into COP tasks.

Firstly, consider Connect-4 on a $6 \times 7$ grid with some pieces already in play. The question is *Can player X force a win in one move?*. This can be converted into a COP task as follows:

```
    def next_empty_row(board, col):
    # next empty row for a particular col
    pass
```

```
def check_win(board,row,col,player):
    # brute force check horizontal, vertical, diagonal for 4 in a row
    pass

board = # set up initial board position

for col in range(7):
    row = next_empty_row(board,col)
    board[row][col]="X"
    if check_win(board,row,col,"X"):
        print("Winning move in column",col)
    board[row][col]=None
```

This pseudocode was written by the authors. Secondly, consider a chess-based task: *Assuming optimal behavior according to the Syzygy tablebase, given a board state X, what is the best move?*. The Syzygy Chess engdame tablebase contains exact game-theoretic results of every legal position with up to 7 pieces. This can be written in COP-style as follows:

```
import chess, chess.syzygy

tb = chess.syzyg.open_tablebases("path/to/syzygy")

# some board state
board = chess.Board("8/8/8/8/8/8/1k6/K5R1 w - - 0 1")
best = max(board.legal_moves, key=lambda m: tb.probe_wdl(board.copy(stack=False)
print(best)
```

Whilst our experiments excluded the use of external libraries, in this instance using the chess library would afford a wide option of chess-based tasks to be converted to COP (best move, set of legal moves, etc.).

## D  QUESTION CLUSTERING

For two questions $i, j$ we compute a similarity distance $d_{ij} = 1 - d_{ij}^{\cos}$. We wish to determine a reasonable threshold distance $d_{\text{thresh}}$ which separates "similar" and "sufficiently different" questions. To determine a value for this threshold, we measure the distances $d_{ij}$ between a set of 2,977 questions generated during our experimentation by three models (Gemini 2.0 Flash, Sonnet 3.5 and GPT-4o), see Figure 7. We experiment with various percentile cut-offs, and observe empirically that a value of $d_{\text{thresh}} = 0.336$ creates good question separation. See Figure 8 for two examples of distinct "question niches", as determined by this cut-off distance.

## E  AUGMENTATION STRATEGIES

In this section we discuss various *augmentation strategies*: prompting frameworks designed to aid LLMs in playing the game optimally. Our framework is designed to be scalable and general, and so we offer a handful of possible strategies, but leave open the prospect of model-specific augmentations.

We present a high-level summary of these strategies in Table 1.

Note that in all strategies the game context is shuffled at inference to create maximal variability in generated questions. Each question is labelled with a question number before shuffling so that ordering information is maintained in the prompt.

In this section we present extra results from the experiment detailed in Section 5. We run a full game of SKATE with four models, but in each experiment equip each model with a different augmentation strategy. In Figure 9 we plot the cumulative average p(correct) scores for each model equipped with

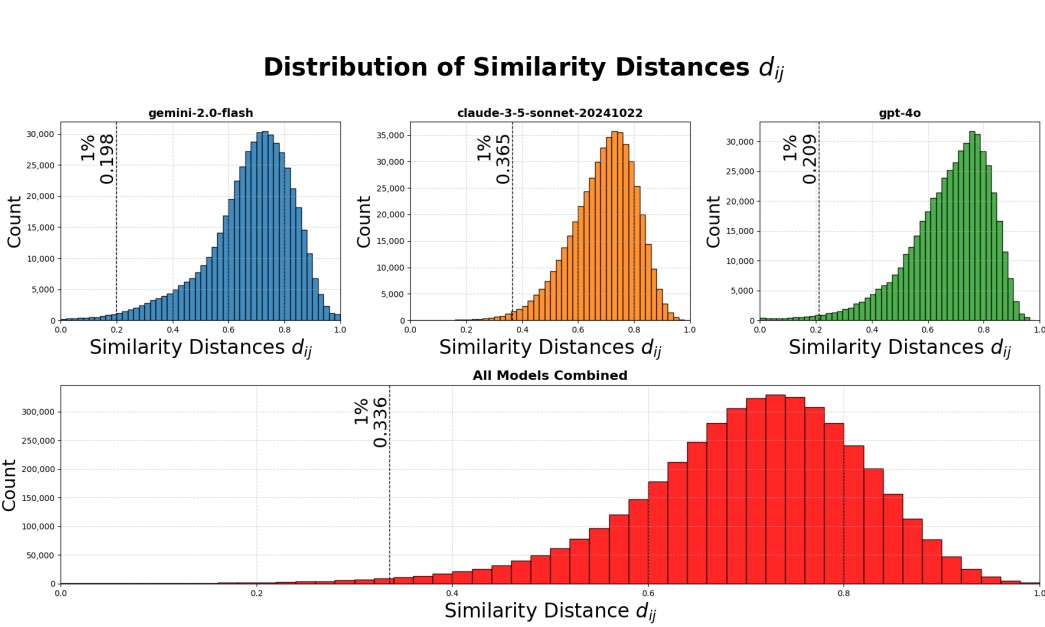

Figure 7: Distribution of similarity distances $d_{ij}$ for a dataset of 3,000 questions.

**Cluster 1**

```
s = "Hello"
t = [ord(c) for c in s]
u = [x-96 if x>96 else x-64 for x in t]
v = sum([i*x for i, x in enumerate(u,1)])
print(v)
```

```
s = "python"
t = {c:s.count(c) for c in s}
u = sorted(t.items(), key=lambda x: (-x[1],x[0]))
print(u[0][0])
```

```
s = "hello"
t = {i:ord(c) for i,c in enumerate(s)}
u = sum(k for k,v in t.items() if v > 108)
print(u)
```

**Cluster 2**

```
global_string = ""

def recursive_function(n):
    global global_string
    if n == 0:
        return
    if n % 3 == 0:
        global_string += str(n)
    else:
        global_string = str(n) + global_string
    recursive_function(n - 1)

recursive_function(7)
print(global_string)
```

```
global_var = 0

def recursive_func(s):
    global global_var
    global_var += 1
    if len(s) <= 1:
        return s
    elif s[0] == s[-1]:
        return s[0] + recursive_func(s[1:-1]) + s[-1]
    else:
        return recursive_func(s[1:]) + recursive_func(s[:-1])

result = recursive_func("abccba")
print(result)
```

Figure 8: Examples of two sets of question clusters, computed using the 0.336 distance threshold.

Table 1: Augmentation strategies summary

| Strategy | Description |
| --- | --- |
| No-Info Baseline | Players are given **none** of the game state information. |
| Historical Tasks | Players are only given **the set of questions they have asked so far** in their context. |
| Historical Performance | Each player is given the set of questions **they have asked so far**, and **their p(correct) scores for those questions** in their context. |
| Full Personal Context | Each player is given the set of questions **they have asked so far**, and **all players' p(correct) scores on those questions** in their context. |
| Full Context | The players are given **all questions in the game archive** and **the p(correct) scores of all players** in their context. |

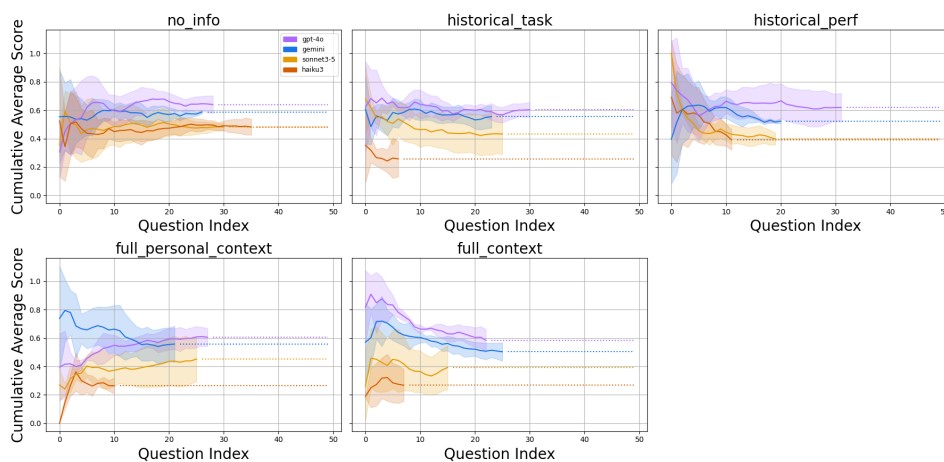

Figure 9: The cumulative average p(correct) score against match steps. We run a full game with four models, but in each experiment equip each model with a different augmentation strategy.

each augmentation strategy. We notice small changes in effect across the strategies. In particular, strategies that include p(correct) data (the last three) suggest that models are adapting their question difficulty throughout the game.

## F    RANKING IS STABLE TO ADDING NEW MODELS

In Figure 10 we show that adding Sonnet 3.5 and then Sonnet 4 (and vice versa) preserves the relative TrueSkill score values of the existing models.

## G    POINTS OF DIFFERENCE

In Figure 11 we plot the histogram of p(correct) variance on each question in the Game of SKATE with 6 players. We give examples of questions with zero variance, in this case either all models score 1.0 or all models score 0.0, as well as two examples with higher variance. These high-variance questions expose clear points of differentiation between models.

## H    ANSWERING V. ASKING SKILL

In Figure 12 we compare two metrics for model capabilities measured on the database of questions obtained from an iteration of our peer-challenge game:

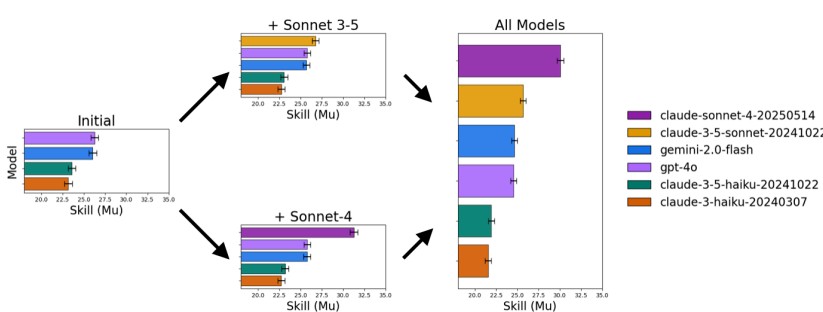

Figure 10: The ranking system is stable to adding new models. On the left-hand plot we begin with the $(\mu, \sigma)$ values for four models. Following the path up and to the right, we first add Sonnet-3.5, then Sonnet-4. Following the path down and to the right, we add the models in the reversed order. We observe that the model ordering is preserved across the two paths.

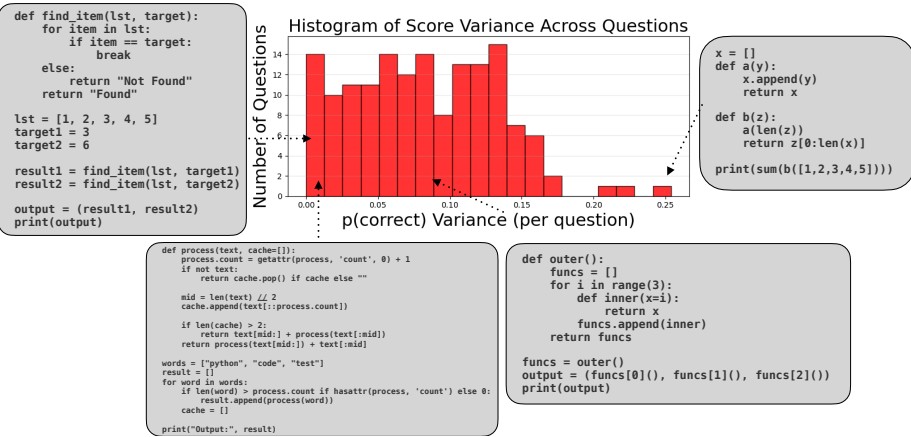

Figure 11: Histogram of variance of competitor p(correct) values, with example questions. All competitors score perfectly on the left-most question (p(correct)= 1), whilst all competitors get p(correct)= 0 on the second question. The third and fourth questions have larger variance in p(correct) values. In the third question, the p(correct)'s are: Sonnet $4 = 1.0$, Sonnet $3.5 = 0.95$, Gemini-2.0-Flash= $0.9$, Haiku $3 = 0.45$, Haiku $3.5 = 0.54$ and GPT-4o $= 1$. In the fourth question Sonnet 4 and Gemini-2.0-Flash score 1.0, whilst *all* other models score $< 0.05$.

1. **Answering Skill:** the average p(correct) score by a subject model on all questions set by competitors.

2. **Asking Skill:** 1 minus the average p(correct) score by all competitor models on the questions set by the subject model.

Figure 12: Answering skill (average p(correct) by each model on other models' questions) v Asking skill (1-average p(correct) by other models on own questions). Sonnet-4's ability is skewed slightly in favour of answering questions, whilst Haiku 3.5's ability leans the other way. The other four models lie closer to the $y = x$ line, but demonstrate performance separation which aligns with their respective TrueSkill rankings in Figure 1.

## I    COMPARISON TO OTHER BENCHMARKS

In Table 2 we compare SKATE TrueSkill $\mu$ scores to published performance on established benchmarks. SKATE scores from Figure 1 (hyperparameters as in main text); other benchmark data compiled from Hugging Face and artificialanalysis.ai.

We see that SKATE correlates well with MMLU-Pro (Wang et al., 2024b) and GPQA Diamond (Rein et al., 2023), but more weakly with Humanity's Last Exam (Phan et al., 2025) (perhaps because all models score low on that benchmark, making ranking noisy) and with BigBench Hard (Zhuo et al., 2025). While we are surprised that SKATE correlates less strongly with BigBench Hard, since we expect broad generalisation of LLM capabilities, we note that SKATE is intended to dynamically explore a broad and 'jagged' capability frontier, rather than to emulate any particular benchmark.

## J    PROMPT DETAILS

All other prompts used in this work are included in the codebase in the Supplementary Material. As an example, below is the prompt used for the SKATE task-setting using the **historical performance** strategy.:

**Generate one novel and challenging Python output prediction puzzle.**

Table 2: Comparison of SKATE TrueSkill $\mu$ scores against published performance on established benchmarks. SKATE scores from Figure 1; other benchmark data compiled from Hugging Face and artificialanalysis.ai. Ranks are given in brackets.

| Model | SKATE (rank) | MMLU-Pro (rank) | HLE (rank) | GPQA Diamond (rank) | BigBench Hard (rank) |
|---|---|---|---|---|---|
| Sonnet 4 | 30.7 (1) | 83.7 (1) | 4.0 (2) | 68.3 (1) | – |
| Sonnet 3.5 2024-10-22 | 25.7 (2) | 78.0 (2) | 3.9 (3) | 59.9 (3) | 30.4 (2) |
| Gemini 2.0 Flash | 24.7 (3) | 77.9 (3) | 5.3 (1) | 62.3 (2) | 28.7 (4) |
| GPT-4o 2024-11-20 | 24.1 (4) | 77.9 (4) | 3.3 (5) | 54.3 (4) | 31.1 (1) |
| Haiku 3 | 21.8 (5) | 42.3 (6) | – | – | 8.2 (5) |
| Haiku 3.5 | 21.7 (6) | 62.1 (5) | 3.5 (4) | 40.8 (5) | 30.1 (3) |
| **Correlation Analysis** | | | | | |
| Pearson correlation | – | 0.74 | 0.18 | 0.87 | 0.55 |
| Spearman rank correlation | – | 0.92 | 0.58 | 0.90 | 0.30 |

**Goal:**

Create a Python code snippet with a single, deterministic output. The puzzle should be maximally difficult for an opponent to predict, yet you must be able to predict its output accurately at least 55% of the time.

**Game Context:**

- Round: {round number} of 50

- Scoring: +1 point for a sufficiently novel question; +1 point for each question answered correctly (55%+ accuracy)

**Strategic Imperatives:**

- **Originality and Variety:** Each round requires a *new* and *highly varied* puzzle. Avoid repetition.

- **Push the Limits:** Design the hardest possible puzzle you believe you can reliably answer.

- The way to win is if your competitor **cannot answer** your questions (and therefore earns no points).

**Instructions for Code Generation:**

1. **Single Python Program:** Generate *exactly one* complete Python code snippet.

2. **Difficulty and Self-Solvability:** The problem *must* be exceptionally difficult for a competitor to solve, yet you *must* be confident in your ability to predict its output accurately.

3. **Maximize Creativity and Subtlety:** Avoid trivial syntax errors or basic variable tracking. Focus on *nuanced* puzzles.

4. **Targeted Reasoning Skill:** Identify and explicitly state the specific Python concept(s) or execution flow detail(s) your code exploits. This will form your "Question Number" explanation.

5. **Deterministic Output:** The code *must* produce a single, predictable output every time. Avoid randomness, user input, or system-dependent behavior.

6. **Built-In Functionality Only:** Restrict code to standard Python 3 built-ins. No external libraries.

7. **Error-Free Execution:** The generated Python 3 code must run without any errors.

8. **Value-Based Output:** The final output should be a specific value (e.g., integer, string, list, tuple, dictionary, or custom object), not a boolean (True/False).

**Uniqueness Constraint:**
Your generated question must be significantly different from all questions you have asked before.

**Failed Attempts**
You only have {num attempts left} in this round to generate a sufficiently unique, verifiable question. Below are your previous failed attempts and why they failed. Learn from your mistakes, and **do not repeat them**.
{failed attempts}

**Archive**
Your generated question must be significantly different from all questions in the provided archive. Furthermore, for each question you can see the score (as a percentage) that you achieved. Use this to calibrate your next question. If you got more than 80% on a question - it is probably **too easy** for your competitor too. Think carefully and write a question which is **harder** than the examples you got high scores on. You will only win the game if you write questions which are so difficult that your competitor will get it wrong. If you got less than 55% on any questions, those types of question are too hard. Go step by step through the examples and use them to guide how hard your next question should be. You should aim for questions which you will score between 60 and 80% on.
{previous questions and p(correct) scores}

**Output Format (Strict Adherence Required):**

- Do *not* include any explanations, external text, or markdown formatting (e.g., triple backticks or quotes).

- The output format must be exact:

  # [short explanation of the specific Python behavior/reasoning skill this question tests]
  <the actual Python 3 code here, no quotes, no markdown, just valid Python code>

## K  INFERENCE COSTS

In this section we give an estimate for the costs involved for running a game of SKATE for N players. For convenience, we quote all numbers of tokens using the OpenAI tokenizer. In the experiments in this paper, using the historical performance augmentation strategy, the number of total number of input tokens for each Asking model was $\sim 130,000$. This number depends on the size of the questions asked by each Asking model, and therefore could vary significantly from model to model. If all models set 50 valid and unique questions, then each model must answer $50(N-1)$ questions. We observe that the average question was of length $\sim 150$ tokens, meaning question answering involves $\sim 7500(N-1)$ input tokens. Estimating output tokens depends on whether or not the models support reasoning and if chain-of-thought is used.

All experiments in this paper cost less than \$500.

## L  USEAGE OF LARGE LANGUAGE MODELS

We acknowledge the use of large language models for minor polishing tasks during the preparation of this paper.

