# OpenReview forum: "SKATE, a Scalable Tournament Eval: Weaker LLMs differentiate between stronger ones using verifiable challenges"
_ICLR.cc/2026/Conference — Submitted to ICLR 2026_

### Official Review · Reviewer_oN5s · 2025-10-28

**Soundness:** 2
**Presentation:** 2
**Contribution:** 2
**Rating:** 4
**Confidence:** 3

**Summary:**

The paper introduces a judge-free evaluation where models set and solve verifiable tasks (i.e., code output prediction MCQs) and are then ranked with TrueSkill while controlling option/order noise and near duplicate questions. Experiments show stable rankings, that weak models can consistently score stronger ones, and demonstrate measurable self-preference when models author questions that they can answer.

**Strengths:**

1) The peer-generated, verifiable tournament evaluation framework is an interesting approach towards a scalable and (hopefully reliable) evaluation framework.
2) The work carefully controls MCQ option/order noise with re-sampling and convergence criteria, adds guardrails that reduce reward hacking, and attempts to enforce question uniqueness via embedding based clustering.

**Weaknesses:**

1) The results are limited to COP. The framework would be stronger with at least one additional verifiable task family.
2) It seems like only one embedding model is used for uniqueness filtering (my apologies if I am mistaken). What happens if a different model is used? How robust is uniqueness filtering to the choice of embedding model?
3) Rankings may be dependent on TrueSkill mapping (eg relative vs absolute), p-threshold, number of distractors, number of rounds, etc.

**Questions:**

1) Would it be possible to add a second verifiable task family?

---

> ### Author Response · Authors · 2025-11-20
> **Response to reviewer oN5s**
>
> We thank the reviewer for their thoughtful comments and feedback. We are pleased by their acknowledgement of the significance of our framework’s scalability, and of the methods we employed to control for MCQ option noise and question redundancy.
>
> _In response to weakness 1 and question 1:_
>
> We thank the reviewer for this point about the limitation of only considering COP tasks. We have added comments and several examples in Appendices B and C on different types of COP tasks and different types of verifiable tasks. We believe that COP is highly generalisable and offers a broad demonstration of SKATE’s power. In particular, in the new Appendix C we describe a broad range of task types which can be framed as COP problems, including algorithmic problem-solving, mathematical reasoning, spatial reasoning, and game playing, amongst others. We hope that these will convince the reviewer that our COP results already encompass a broad range of possible task types.
>
> In a new Appendix B we give examples of alternate types of verifiable tasks (including mathematical proofs and code generation) which cannot be solved with access to code execution tools. It would be interesting for future work to consider running new SKATE experiments with these task types.
>
> _In response to weakness 2:_
>
> The reviewer is correct, our current method uses a single embedding model for uniqueness filtering. While it’s possible that different embedding models could lead to different filtering decisions, we expect the overall impact of this to be minimal for current models, especially since in practice the filtering step mostly removes fairly obvious near-duplicates. We have updated the manuscript to mention this.
>
> _In response to weakness 3:_
>
> The reviewer is correct that there are a number of hyperparameters which could be explored. Some of these we mention in the paper (e.g. Figure 2 shows both absolute and relative TrueSkill mapping), some we don’t explore but expect minimal effect (e.g. 3 distractors from a set of 9 rather than 5 from 20), and some of which we show the approach is stable to (e.g. Figure 1 shows convergence over sufficient number of rounds).

---

### Official Review · Reviewer_JWwY · 2025-10-31

**Soundness:** 3
**Presentation:** 3
**Contribution:** 3
**Rating:** 8
**Confidence:** 4

**Summary:**

The paper introduces a methodology for LLMs collaboratively and interactively judge each other's performance, by having the LLMs play a game wherein they generate questions whose answers can be verifiably answered. The aim is to pose questions they can answer while the other LLMs cannot distinguish between true and incorrect answers. The LLMs are ranked by their abilities to both correctly answer other LLM's questions, and to stump the other LLMs.

**Strengths:**

- the evaluations are verifiable, differing from common LLM-as-Judge frameworks like Alpaca-Eval, where the biases of a judge LLM influences the evaluation
- no human input is required in generating the evaluation data sets
- the resulting rankings (on code output prediction) are demonstrated to be stable to the addition of more LLMs
- the methdology is shown to elicit some fine-grained performance differences between LLMs,. and incorporate varying levels of prior knowledge to assist the LLMs in generating difficult questions
The primary significance is in the verifiability of the methodology.

**Weaknesses:**

the methodology is limited to automatedly verifiable tasks, while many tasks we would like to understand the performance of LLMs on (e.g. alignment), cannot be automatically verifiable
- the methodology similarly is limited to tasks that can be posed as multiple choice questions, so it cannot be used to e.g. compare the summarization abilities of LLMs
- although the methodology encourages diversity, it does not ensure coverage, so e.g. in the specific case studied in the paper, it could be the case that LLMs all share a common blind spot for code output prediction that is not identified by this methodology.
- this methodology was only instantiated and tested on a code output prediction task; it would have been more persuasive to see it perform on at least one other, significantly different task

**Questions:**

- please address the statistical issues in using adaptive sampling to determine p(correct): this means that models with less certainty are given a larger budget of samples.
- please be more comprehensive with your discussion of how your work compares to related gamified LLM-vs-LLM evaluation methodologies, e.g. the GTBench paper of Duan et al. in NeurIPS 2024 and the ZeroSumEval paper of Alyahya et al. in ACL 2025; in particular, how does your use and emphasis placed on verifiability differ from theirs? Making this more clear would help me better judge the novelty and significance of your contribution.
- the author list in the bibliographic entry for Humanity's Last Exam spans several pages, it should be shortened! Several other author lists should be similarly be shortened.

---

> ### Author Response · Authors · 2025-11-20
>
> We sincerely thank the reviewer for the positive assessment of our work's contribution and methodology. Your feedback is highly valuable for strengthening our paper. Below, we address the concerns in detail.
>
> _In response to weakness 1:_
>
> The reviewer makes a valid point about multiple-choice questions. We have updated the paper to address this in both Section 3 and Appendices B and C. We restricted our experiments to multiple-choice questions, but in fact SKATE applies to open-ended questions also: the key requirement is that it must be possible to automatically verify the correctness of a question / task. For example, in Appendix B we describe how SKATE could incorporate two such tasks: theorem-proving in a formal language, and test-driven code development. The reviewer is still correct that our approach is not completely general, but this is only due to our verifiable requirement: for example, for text summarization tasks it would be more difficult to formally verify the correctness of a given solution.
>
> _In response to weakness 2:_
>
> This is another valid limitation of our work: while the similarity metric we deploy ensures that some exploration is achieved, we do not have any guarantees about full coverage (and indeed expect current task-setting models to be far from attaining this). An interesting direction for future work, with the aim of eliciting more exploration, would be to “warm start” task-setters’ search with seed questions. These questions could be generated across a spectrum of domains and by a range of models. We have added a comment about this to the Discussion Section.
>
> _In response to weakness 3:_
>
> We thank the reviewer for this point. We have added comments and several examples in Appendix B and C on both different types of COP tasks and different types of verifiable tasks.
>
> _In response to question 1:_
>
> Thank you for this point. Originally we gave all models the same budget when running our experiments and found that in many instances the models’ scores converged quickly, and error bars no longer significantly differed with increasing sampling. As a result we implemented our early stopping algorithm (Algorithm 1) to avoid unnecessary budget spend. We would appreciate any feedback if the reviewer can see a failure mode of this approach.
>
> _In response to question 2:_
>
> We thank the reviewer for pointing out these references and have included a paragraph on other LLM-v-LLM settings in the Related work section. We believe the key distinction between these works and ours is that in SKATE LLMs both pose and answer questions. GTBench has static game-theoretic environments, and ZeroSumEval has 7 fixed types of game which LLMs play against one another. In both cases the tasks are human-created. In SKATE the challenges posed are dynamic and varying depending on the strategy of the LLMs playing - meaning our method naturally scales as models improve, to test a wider range of capabilities.
>
> _In response to question 3:_
>
> We agree that the author list for HLE is extremely long, and would personally prefer to abridge it. However, we are using the ICLR LaTeX template which has clear and explicit instructions not to alter such formatting.

---

### Official Review · Reviewer_3DNS · 2025-10-31

**Soundness:** 2
**Presentation:** 3
**Contribution:** 2
**Rating:** 2
**Confidence:** 4

**Summary:**

This paper introduces SKATE, a novel LLM evaluation framework where models compete by generating and solving verifiable tasks for one another. Models act as both "task-setters" and "solvers" in a tournament format, creating Code-Output-Prediction (COP) challenges. The system uses TrueSkill ranking and tests 6 frontier LLMs, finding that: (1) weaker models can differentiate stronger ones, (2) models exhibit self-preferencing in question generation, and (3) automatic discovery of differentiating questions.

**Strengths:**

1. The whole idea is novel and well-motivated.The paper identifies two major bottlenecks in LLM evaluation: (1) evaluation requires costly, non-scalable human-annotated ground truths, or (2) relies on LLM-as-judge which is easily manipulated. SKATE attempts to address both through a peer-challenge multi-model system.
2. Methodologies are sound and considerate, such as robust scoring algorithm to adress the multiple-choice biases and question clustering to increase the diversity of questions.

**Weaknesses:**

1. While the authors claim that COP tasks provide "a general substrate for evaluating model capabilities," this paper provides zero empirical evidence that SKATE can work with other task types. The COP tasks tested here don't even resemble a typical coding evaluation where models' generation or algorithmic problem-solving capabilities are tested. Any model with code execution tools could solve COP tasks. Without demonstrating SKATE generalisation ability on more diverse verifiable tasks (such as mathematical proofs, puzzles), the claim is questionable.

2. Although COP tasks can be verified through code execution easily, the authors provide no evidence that general verifiable tasks can have similar guarantees. For instance, if a model generates a mathematical problem as a challenge (the task itself is verifiable with ground-truths but extremely hard with only models), verifying the correctness of the answer may be non-trivial or even intractable.

3. While collective knowledge from multiple models works well for training scenarios, using it for evaluation may be problematic because it provides no ground-truth anchor—evaluation results depend entirely on which specific models are included in the cohort. The experiments in Section 6.2 only test adding models sequentially by capability order, which is insufficient to establish robustness. A rigorous validation would require testing with diverse model combinations to show rankings remain consistent, or anchoring the evaluation with human-verified questions to provide external validation.

4. The empirical findings presented in Section 6, while technically sound, lack novelty and offer limited new insights into LLM capabilities. The observation that weaker models can reliably score stronger ones, and models prefer their own questions, have been found two years ago.

**Questions:**

See weakness.

---

> ### Author Response · Authors · 2025-11-20
>
> We are pleased that the reviewer considers the overall idea to be novel and well-motivated, and agrees with the identification of the two major bottlenecks in LLM evaluation which this paper aims to address. We also appreciate the description of our methodologies as “sound and considerate”.
>
> _In response to point 1:_
>
> We thank the reviewer for making the point that our original version of the paper insufficiently supported the claim that Code-Output Prediction (COP) can be generalised to many different task types. We see the generalisability of COP as a major strength of the paper, and have sought to clarify our findings and claims in our updates to the paper. To this end, we have added Appendix C to the manuscript, detailing a broad range of task types which fit within the scope of COP. Our claim is that COP is highly flexible and that many types of tasks can be re-written in this way (in the appendix we give examples of mathematical reasoning, spatial reasoning, and game play).
>
> The reviewer is correct that models with access to code execution tools could trivially solve COP. We limit the experiments in this run of SKATE to models without this ability. We still consider COP a meaningful test of model capabilities, similar to how “mental arithmetic” is a meaningful test of human cognition even in the age of calculators.
>
> In a new Appendix B we give examples of alternate types of verifiable tasks (including mathematical proofs and code generation) which cannot be solved with access to code execution tools. We think these would be interesting avenues for future work.
>
>
> _In response to point 2:_
>
> The reviewer’s second point is also important: that some tasks may be verifiable in-principle, but this does not necessarily mean that their “ground truth” can be easily or automatically verified. We have clarified in Section 3 of the manuscript that SKATE is designed for automatically verifiable tasks, though other tasks may exist which are harder to verify. For example, in a new Appendix B we have described one way that mathematical problem-solving does in fact still work within our framework: when written in a formal language (e.g. Lean). In this setting, models can pose theorems in Lean, and answers are submitted in Lean. The formal language allows for automatic verification of the solution.
>
> _In response to point 3:_
>
> We’d like to point out that in Figure 10 we experiment with adding models in diverse orders (not just sequentially as noted in the reviewer’s comment), and we hope this addresses the concern pointed out here.
>
> _In response to point 4:_
>
> We thank the reviewer for the connection of our findings to existing literature, and we would welcome any references which we could include for context. We consider the object-level insights into model capabilities to be a secondary result of this work, and the main contribution to be the scalable and automated framework itself. As such, we consider any agreement with existing literature to actually be a strength, validating the SKATE framework. We hope this framework can surface further such insights automatically, as increasingly capable models are developed.

---

> ### Comment · Reviewer_3DNS · 2025-11-25
> **Official Comment by Reviewer 3DNS**
>
> Thanks for the detailed response from the authors. I think the supplementary experiments review that SKATE can be applied to other verifiable tasks, like COP and Lean, but under the condition that the environment itself is verifiable [1]. In the words, SKATE can only be tested on a limited domain. This makes SKATE a more conceptual idea rather than something we can really deploy in the real world. Thus I would only like to raise my score to 4.
>
> [1] Zeng Z, Ivison H, Wang Y, et al. RLVE: Scaling Up Reinforcement Learning for Language Models with Adaptive Verifiable Environments[J]. arXiv preprint arXiv:2511.07317, 2025.

---

### Author Response · Authors · 2025-11-20
**Joint Response to All Reviewers**

We thank all three reviewers for their thoughtful and constructive feedback. We are pleased that the core motivations and methodological contributions came through clearly. In particular, we appreciate the reviewers’ recognition that our approach directly addresses major limitations of current evaluation methods: (i) the lack of scalability arising from the need for significant human involvement in question generation and, (ii) the well-documented challenges of LLM-as-judge frameworks.

__Key strengths noted by the reviewers:__

Across reviews, multiple strengths of our contribution were highlighted. We thank R1 for noting that our work identifies two central bottlenecks in LLM evaluation and effectively addresses them through our peer-challenge framework. We also appreciate their positive assessment of our methodologies as “sound and considerate”, as well as their recognition that our clustering procedure supports more robust results. We are grateful for R2 for underscoring the importance of verifiability of our method and for highlighting that our pipeline surfaces interesting fine-grained capability differences between LLMs. We also thank R3 for recognizing scalability as a key contribution and for commending the care we took in handling multiple-choice noise and question clustering.

These strengths affirm our work’s main aim to provide a __scalable, verifiable and robust__ framework for evaluating LLMs without relying on human-data or subjective model judgements. We want to thank the reviewers again for their careful assessment and helpful suggestions.

__Key revisions in response to the comments:__

We have revised the paper carefully, with major changes summarized below.
Both R1 and R3 raised concerns regarding our choice of code-output-prediction (COP) as the primary class of verifiable tasks used to demonstrate SKATE. We agree that our original presentation did not sufficiently emphasize the generality of COP-style questions. In response, we have added a new Appendix C (“Comments on Generalizability of COP”), which illustrates how a broad range of tasks such as mathematical reasoning, game-playing and spatial reasoning, can all be expressed in COP format. The majority of the examples included in this Appendix were written by LLMs: emphasizing that these tasks are a) representable as COP, and b) possible to generate automatically with current models.

R1 also asked about the applicability of our results to other verifiable tasks. We appreciate this point and have clarified our claim in Section 3. Specifically, we argue that SKATE applies to all __automatically verifiable tasks__, meaning tasks whose solutions can be checked automatically even if an LLM cannot always generate the ground truth. We provide further discussion and concrete examples in a new Appendix B (“Comments on Other Verifiable Tasks”).

We also thank R2 for the comments on related work and formatting; these have been addressed in detail in the individual reviewer responses. We have also addressed R2’s point about whether or not our method was limited to multiple-choice questions. We restricted our experiments to this class of task, but in fact SKATE applies to open-ended questions also.  The key requirement is that it must be possible to automatically verify the correctness of a question / task. We include examples in Appendix B of open-ended questions which fit this requirement and discuss how models should be scored in this setting.

[In this comment, we refer to Reviewer 3DNS as R1, Reviewer JWwY as R2, and Reviewer oN5s as R3.]

---

### Author Response · Authors · 2025-12-04
**Note to AC**

Due to the updated rebuttal process, here we have included a short comment for the AC. We believe that the positive and constructive feedback offered has helped us clarify and improve important aspects of our work.

After responding in detail to all three reviewers, unfortunately we only received a reply from one (3DNS) before the leak. We’d like to highlight that 3DNS decided to update their score to a 4 before the leak, and we believe we have thoroughly addressed all concerns raised by the other reviewers in the individual responses.

Please see “Joint Response to All Reviewers” for a high level summary of these responses.

---

### Meta-Review · Area_Chair_UVti · 2025-12-10

**Summary:**

There is a clear mismatch between the framing and stated goal of the paper and its execution.

**Reviewer Concerns:**

Overall, while aiming for models to be testing against each other, acting as both tester and testee (as opposed to using [TextArena.ai](https://github.com/LeonGuertler/TextArena) style games, where the environment is fixed but the difficulty changes because of the skill of the opponent). Across reviewers, there is an ask to add more proof that the method is indeed general. Personally, I believe this is an exciting idea, but not a general one, at least not as general as the authors propose. Instead, any place where an open-domain challenge can be created, in a way where the answer can be verified by non LLM result is where such methods could be employed. Whether there are many of those or this can be generalized to other cases (without the model cheating about the answer they expect) is up to the authors to show, or to focus their claims on what they show.

**Reviewer Scores:**

That is not a fair, relevant or meaningful question. I protest the way this was all handled.
A Reviewers are not here, and ToM is weak, at least mine and the one literature study. I will not try to predict people.
B Scores are, anyway, a weak signal of interest; a paper should not be accepted or rejected just based on it. An AC's job is to look at the specific weaknesses and translate them into a recommendation.
C There are about 100 pages of discussions for me to read overall, in addition to the discussions I monitored and were just replaced, this is beyond my personal ability to do fairly. I did my best effort.

---

### Decision · Program_Chairs · 2026-01-26

Reject